Extraordinary incidence of cervical ribs indicates vulnerable condition in Late Pleistocene mammoths

Reumer Jelle W.F. 1 2
ten Broek Clara M.A. 3 4
Galis Frietson 3 frietson.galis@naturalis.nl
1 Natural History Museum , Rotterdam , The Netherlands
2 Faculty of Geosciences, Utrecht University , Utrecht , The Netherlands
3 Naturalis Biodiversity Center, Terrestrial Zoology/Geology , Leiden , The Netherlands
4 University Antwerp, Evolutionary Ecology Group , Antwerp , Belgium
Farke Andrew
Electronic publication date: 2014 Mar 25
Publication date: 2014
Volume: 2
Electronic Location ID: e318
Received 2014 Feb 4; Accepted 2014 Mar 3
Copyright: © 2014 Reumer et al.
Copyright year: 2014
Copyright holder: Reumer et al.
License: This is an open access article distributed under the terms of the Creative Commons Attribution License, which permits unrestricted use, distribution, reproduction and adaptation in any medium and for any purpose provided that it is properly attributed. For attribution, the original author(s), title, publication source (PeerJ) and either DOI or URL of the article must be cited.
License URL: https://creativecommons.org/licenses/by/4.0/

Keywords: Mammoths, Extinction, Loxodonta, Elephas, Vertebral column, Body plan, Inbreeding

Funding: Royal Museum for Central Africa Tervuren Zoological Museum Copenhagen Natural History Museum of Stockholm BE-TAF-1649 DK-TAF-2183 SE-TAF-3009 FG acknowledges Synthesys travel grants to visit the Royal Museum for Central Africa Tervuren, the Zoological Museum Copenhagen, and the Natural History Museum of Stockholm (BE-TAF-1649, DK-TAF-2183, DE-TAF-2114, SE-TAF-3009). The funders had no role in study design, data collection and analysis, decision to publish, or preparation of the manuscript.

==============================
The number of cervical vertebrae in mammals is highly conserved at seven. We have shown that changes of this number are selected against due to a coupling with major congenital abnormalities (pleiotropic effects). Here we show that the incidence of abnormal cervical vertebral numbers in Late Pleistocene mammoths from the North Sea is high (33.3%) and approximately 10 times higher than that of extant elephants (3.6%). Abnormal numbers were due to the presence of large cervical ribs on the seventh vertebra, which we deduced from the presence of rib articulation facets on sixth (posterior side) and seventh (anterior side) cervical vertebrae. The incidence of abnormal cervical vertebral numbers in mammoths appears to be much higher than in other mammalian species, apart from exceptional sloths, manatees and dugongs and indicates a vulnerable condition. We argue that the increased incidence of cervical ribs in mammoths is probably caused by inbreeding and adverse conditions that impact early pregnancies in declining populations close to extinction in the Late Pleistocene.

Introduction

The number of cervical vertebrae in mammals is remarkably constant at seven. In other tetrapods, the number of cervical vertebrae varies considerably, and in mammals the number of vertebrae in more caudal vertebral regions is variable as well (Leboucq, 1898; Schultz, 1961; Starck, 1979; Narita & Kuratani, 2005). Only manatees (Trichechus, Sirenia) and sloths (Bradypus and Choloepus, Xenarthra) have an exceptional number of cervical vertebrae (Bateson, 1894; Starck, 1979; Varela-Lasheras et al., 2011). Despite the extreme evolutionary conservation of the number of cervical vertebrae, intraspecific variation is not uncommon in mammals. The most common variation is represented by ribs on the seventh vertebra, so-called cervical ribs, which can be considered a partial or complete homeotic transformation of a cervical into a thoracic vertebra (involving a change in the activity of Hox genes (Galis, 1999; Li & Shiota, 2000; Varela-Lasheras et al., 2011; Wéry et al., 2003)). The strong conservation of the number of cervical vertebrae implies that there must be selection against intraspecific variation of this number. Indeed, very strong selection against cervical ribs was shown to exist in humans (Galis, 1999; Galis et al., 2006; Furtado et al., 2011; ten Broek et al., 2012). Approximately 90 percent of individuals possessing a cervical rib die before reaching reproductive age (Galis et al., 2006). The severe selection is due to the strong association of cervical ribs with multiple and major congenital abnormalities. In other mammalian species, we have also found an association with abnormalities (Varela-Lasheras et al., 2011). A cervical rib itself is relatively harmless, but its development is induced by a (genetic or environmental) disturbance of early embryogenesis (Li & Shiota, 2000; Wéry et al., 2003; Chernoff & Rogers, 2004; Galis et al., 2006). Such a disturbance usually has multiple effects, due to the highly interactive nature of early embryogenesis. Hence, the strong selection against cervical ribs is indirect and due to the severity of the associated medical problems (Galis et al., 2006; ten Broek et al., 2012).

Of three caudal cervical vertebrae from Mammuthus primigenius, a sixth (C6) and two seventh (C7), that were recently found in the North Sea, during infrastructural works for an extension of the Rotterdam Harbour (Maasvlakte 2) and donated to the Natural History Museum in Rotterdam, two possessed articulation facets for cervical ribs (the C6 and one of the C7). This surprising finding aroused our interest, and we searched the extensive collection of Late Pleistocene M. primigenius material in the Naturalis Biodiversity Centre (Leiden) to make an estimate of the incidence of this developmental abnormality. Additionally, we determined for comparison the incidence of cervical ribs in skeletons of the most closely related extant species, the Asian and African elephants (Elephas maximus and Loxodonta africana).

Methods

Specimens

We analyzed 6 sixth cervical vertebrae (C6) and 10 seventh cervical vertebrae (C7) of Late Pleistocene mammoths (M. primigenius), from two collections: the Natural History Museum Rotterdam (NMR, Table 1) and Naturalis Biodiversity Center (Naturalis, Table 1). The cervical vertebrae were identified as C6 and C7 based on the relative size of the spinous processes and anterior tubercles. We analysed 28 specimens of extant elephants, 21 E. maximus and 7 L. africana, from 5 collections: the Natural History Museum of Denmark, Copenhagen (ZMUC), Naturhistorisches Museum Wien, Vienna (NHMW), The University of Vienna, the Swedish Museum of Natural History, Stockholm (NRM), the Royal Museum for Central Africa Tervuren (RMCA) and Naturalis Biodiversity Center (Naturalis). All mammoth specimens (see Table 1 for collection numbers) are of Late Pleistocene age and originate from the North Sea. Two specimens (C6, inv.nr. NMR999100006627 and C7, inv. nr. NMR999100007602) were recently found during infrastructural works in the Rotterdam harbor area (“Maasvlakte 2”) on the North Sea seabed (Maasvlakte Zandwingebied, i.e., the source area for sand extraction, c. 51°59′N/3°53′E) and allocated to the NMR by the Rotterdam Port Authorities.

Table 1 List of investigated specimens and scores of articulation facets of cervical ribs.

The presence of articulation facets of ribs was indicated posteriorly on the sixth cervical vertebra (C6) and/or anteriorly on the seventh cervical vertebra (C7).

Species	Institute	Collection no.	Sex	Vertebra	Rib facets (left/right)	
Mammuthus primigenius	Naturalis	RGM592809	n.a.	C7	0	
RGM103337	n.a.	C7	n.a.	
RGM132902	n.a.	C7	0	
RGM139079	n.a.	C7	0	
RGM172327	n.a.	C7	n.a.	
RGM20026	n.a.	C7	n.a.	
RGM20313	n.a.	C6	n.a.	
RGM369465	n.a.	C6	n.a.	
RGM40098	n.a.	C6	0	
RGM40120	n.a.	C7	0	
RGM4445989	n.a.	C7	n.a.	
RGM79245	n.a.	C6	n.a.	
RGM146248	n.a.	C6	1 (left)	
NMR	NMR999100007602	n.a.	C7	1 (left)	
NMR999100006627	n.a.	C6	1 (right)	
NMR999100007479	n.a.	C7	0	
Elephas maximus	Naturalis	RMNH.MAM.46016	n.a.	C6, C7	0	
RMNH.MAM.46024	M	C6, C7	0	
RMNH.MAM.39235	F	C6, C7	0	
RMNH.MAM.39234	n.a.	C6, C7	0	
ZMA 13483	n.a.	C6, C7	0	
RMNH.MAM.46018	n.a.	C6, C7	0	
ZMA.MAM.30069	M	C6, C7	0	
NRM	A609596*	F	C6, C7	0	
A591540	n.a.	C6, C7	0	
A600572	n.a.	C6, C7	0	
A589489*	F	C6, C7	0	
NMW	16545	n.a.	C6, C7	0	
5505*	M	C6, C7	0	
UAV	n.a.	n.a.	C6, C7	0	
ZMUC	ZMUC CN2	F	C6, C7	0	
ZMUC CN4196*	n.a.	C6, C7	0	
ZMUC CN1399*	F	C6, C7	0	
ZMUC CN1	M	C6, C7	0	
ZMUC CN2293*	M	C6, C7	0	
ZMUC CN639*	F	C6, C7	1 (right, C7)	
ZMUC CN 558*	M	C6, C7	0	
Loxodonta africana	Naturalis	RMNH.MAM.45488	M	C6, C7	0	
NRM	A601286	M	C6, C7	0	
A600551	M	C6, C7	0	
NMW	287* (exhibition)	n.a.	C6, C7	0	
RMCA	RMCA 4559	n.a.	C6, C7	0	
ZMUC	ZMUC CN708*	M	C6, C7	0	
ZMUC CN3684*	M	C6, C7	0	
Notes.

* Died in captivity (wild born).

n.a. not available

Naturalis Naturalis Biodiversity Center Leiden

NRM Naturhistoriska Riksmuseet Stockholm

NMW Naturhistorisches Museum Wien

UAV University Anatomy Vienna

ZMUC Zoologisk Museum University Copenhagen

RMCA Royal Museum Central Africa Tervuren

Cervical ribs

We analyzed the C6 and C7 vertebrae for the presence or absence of articulation facets of cervical ribs. The presence of cervical ribs can be deduced from articulation facets on the anterior side of C7 (Figs. 1A and 1D) and, if the cervical ribs are large enough, on the posterior side of C6, as well (Figs. 1B and 1C).

Figure 1 Presence of rib articulation facets on cervical vertebrae of woolly mammoths (A–C) and Asian elephant (D).

(A) Posterior view of a C6 of a Mammuthus primigenius from the North Sea (NMR999100006627), showing an articulation facet of a cervical rib on the right side. (B) Anterior view of a C7 of a Mammuthus primigenius from the North Sea (NMR999100007602), showing a sinistral articulation facet (lower right in the picture). (C) Posterior view of a C6 of a Mammuthus primigenius from the North Sea (Naturalis St 146248), showing an articulation facet of a cervical rib on the left side. (D) Anterior view of a C7 of an Elephas maximus (ZMUC CN639), showing a minute articulation facet of a cervical rib on the right side (see inset for articulation facet). The size of cervical ribs is presumably associated with the strength of associated abnormalities. Arrows indicate articulation facets.

Statistical tests

To compare the prevalence of cervical rib facets between mammoths and elephants we used a G-test of independence, which is particularly appropriate for variable samples sizes as is often the case with paleopathological data (Farke, 2007). Furthermore we also used Barnard’s test for 2 × 2 tables, which is appropriate for small sample sizes and yields greater power than Fisher’s exact test (Barnard, 1945). P-values <0.05 were considered as significant. All analyses were carried out in R.

Results

Articulation facets for cervical ribs on cervical vertebrae are characterized by the following combination of characteristics: (i) they have a smooth, polished-looking surface, visibly smoother than the (surrounding) cortical surface of the vertebrae; (ii) the surfaces have no vascular or nervous foramina; and (iii) the facets are bordered by a clear edge, distinguishing them from the surrounding cortex.

We found one C7 with a unilateral sinistral anterior rib facet indicating a left cervical rib (Fig. 1A). Five C7 did not have rib facets anteriorly and four could not be judged due to the absence of the relevant part of the vertebra. We found two C6 with rib facets on the posterior side indicating cervical ribs: one on the right side and one on the left side (Figs. 1B and 1C respectively). We found one C6 without rib facets posteriorly and three that could not be judged.

Thus, out of the nine C6 and C7 that could be evaluated, three indicate the presence of a cervical rib, i.e., an incidence of cervical rib facets of 33.3%. We found in one of the 21 E. maximus a minute cervical rib facet on C7 (Fig. 1D, 4.8%) and no articulation facet visible posteriorly on C6 of the same individual. None of the seven L. africana individuals had cervical rib facets, nor were rudimentary cervical ribs found. The overall incidence of cervical ribs in the two species is, thus, 3.6%. This is significantly lower than the 33.3% in mammoths, if we only consider vertebrae that can be evaluated for cervical rib articulation facets (G-test for independence, p = 0.035, Barnard’s exact test, p = 0.031).

Discussion

The incidence of cervical rib facets in our set of Late Pleistocene M. primigenius recovered from the North Sea is extremely high (3 out of 9, 33.3%), almost ten times higher than that of extant elephants (1 out of 29, 3.6%). In humans, an incidence higher than 1% has only been found in hospitals or isolated populations (Galis et al., 2006). An incidence higher than 25% has only been found in children with leukemia, brain tumours and neuroblastoma (Schumacher, Mai & Gutjahr, 1992; Galis & Metz, 2003; Merks et al., 2005) and in deceased fetuses and infants (Galis et al., 2006; Furtado et al., 2011; ten Broek et al., 2012). Along with the high incidence of cervical ribs in mammoths, the size of the articulation facets is particularly large (Figs. 1A–1C), substantially larger than the articulation facet found in the one E. maximus (Fig. 1D) and, pointing to substantially larger cervical ribs than usually found in humans (see Bots et al., 2011; ten Broek et al., 2012 for examples). Size of cervical ribs was found to be negatively correlated with fitness in transgenic mice (Jeannotte et al., 1993; see also Bots et al., 2011).

The exceptionally high incidence of large cervical ribs in our set of Late Pleistocene mammoths can be due to two factors. Firstly, it can be due to a high rate of inbreeding in declining populations, before final extinction. A high incidence of cervical ribs (7.46%) has been observed in an isolated human population (Palma & Carini, 1990) in Sicily, in inbred pedigreed dogs (11.4% Breit & Kunzel, 1998) and inbred minipigs (11% at birth, Jørgensen, 1998). Generally, in inbred mammals there is an increased incidence of congenital anomalies (Cristescu et al., 2009; Räikkönen et al., 2013). Recent studies have shown that the genetic diversity was extremely low in Late Pleistocene mammoth populations in Siberia (Miller et al., 2008; Nyström et al., 2012). Additionally, the increased incidence of cervical ribs may be due to harsh conditions that impact early pregnancies, because diseases, famine, cold and other stressors can lead to disturbances of early organogenesis, that can result in the induction of cervical ribs (e.g., Sawin, 1937; Li & Shiota, 2000; Wéry et al., 2003; Chernoff & Rogers, 2004; Steigenga et al., 2006). Harsh conditions during the Late Pleistocene, a period of intense climatic fluctuations and ecosystem instability, are plausible (Brace et al., 2012). Furthermore, bone dystrophy in mammoth calves of Northern Eurasian Late Pleistocene populations is found regularly and assumed to be caused by mineral deficiencies in pregnant females (Leshchinskiy, 2012). Hence, a combination of inbreeding and harsh conditions may be the most likely explanation for the extremely high incidence of cervical ribs. Our results, thus, are in agreement with inbreeding in populations in North-Western Eurasia, just as has been found for Siberian populations (Miller et al., 2008; Nyström et al., 2012). Finally, the high incidence and large size of the cervical ribs indicates a strong vulnerability, given the association of cervical ribs with diseases and congenital abnormalities in mammals. The vulnerable condition may well have contributed to the eventual extinction of the woolly mammoths.

We thank the Rotterdam Port Authorities for the donation of all bones that are found during the extension of the Rotterdam harbor in the North Sea to the Natural History Museum in Rotterdam. We thank Mogens Andersen, Alex Bibl, Daniela Kalthoff, Steven van der Mije, Wim Wendelen and Reinier van Zelst for making specimens available and Alexandra van der Geer, Jacques van Alphen, Russell Lande, Natasja den Ouden and Rienk de Jong for comments. We thank Joris van Alphen for making the photographs of Figs. 1A–1C.

Additional Information and Declarations

Competing Interests

Author Contributions

The authors declare no competing interests.

Jelle W.F. Reumer and Frietson Galis conceived and designed the experiments, performed the experiments, wrote the paper, prepared figures and/or tables, reviewed drafts of the paper.

Clara M.A. ten Broek performed the experiments, analyzed the data, prepared figures and/or tables, reviewed drafts of the paper.

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
