# Peer review of "Extraordinary incidence of cervical ribs indicates vulnerable condition in Late Pleistocene mammoths"

_PeerJ, doi:10.7717/peerj.318_

## Round 0.1 · original submission · Minor Revisions

1) The suggestions by the reviewers are generally quite reasonable, and should be incorporated as suggested. I have a copy of the manuscript with some additional suggestions and copy-edits, which I will email separately.

2) Reviewer 1 questions the choice of statistical test, and I agree that this should be given some thought. A G-test of independence may be most appropriate here--see Farke 2007 for one explanation of statistical approaches to pathologies and other anomalies (Farke, A. A. 2007. Reexamination of paleopathology in plesiosaurs. Journal of Vertebrate Paleontology 27:724-726.)

Reviewer 1 ·

Basic reporting

The article is consistent with journal policies as I understand them. There are some minor problems with English expression that will be noted in comments below.

Experimental design

The methodology is clearly described. The only procedural question I was left with is why the authors used Barnard’s test rather than Fisher’s Exact test. Because of time limitations, I cannot now pursue this issue further.

Validity of the findings

This is an interesting set of observations. Still, the sample is so small that despite the significant p-value, it is hard to be confident that the effect is biologically significant. See the article by Nuzzo on statistical errors in a recent Nature. With respect to statistics, it also seems it would be relevant to report the actual p-value, not just that it was lower than 0.5.

Additional comments

Additional comment for the authors’ consideration:
1. Lines 88-92: It is certainly possible that large articulation facets might be associated with large ribs, but it is not clear to me that this would necessarily be so. Nor is it clear why the size of cervical ribs would be “negatively correlated with fitness” even if their existence is negatively correlated with fitness.

Editorial recommendations:
In abstract, do not abbreviate approximately as “approx.”
Line 35: insert “the” before “C6”.
Line 36, 43, 50, 93: use lower-case “l” for “late”; time interval is informal.
Line 39: insert “closely” after “most”.
Line 40, 46, 76: use lower-case “a” in “africana”.
Line 76: change “did have” to “had”.
Line 83, 84: use singular “incidence” because the usage is generic, not enumerative. For line 84, “An incidence higher than 25% has only ….”
Line 96: insert space before “(Palma”.
Line 106: change “intensive” to “intense”.
Line 108: delete “a”.
Line 110: change to “explanation for the”.

·

Basic reporting

The manuscript is clearly written and meets the appropriate standard.

In general
Some minor format styles need to be addressed in order to conform to PeerJ style.
In text citations: Multiple references to the same item should be ordered chronologically. Change from alphabetical order to chronological order at lines: 11-12; 18-19; 21-22; 27-28; 85-86; 86-87; 92; 104-105.
Reference section: write full journal name at L144 and L177 so it conforms to the other references. Do not use “and” between author names.

Other formatting: L58, L75 and L88: Point should follow “Fig” or “Figs”. Also, uppercase letters should be used for each figure part in the figure text to Figure 1 so it conforms to PeerJ style.

The abstract states:
“Here we show that the incidence of abnormal cervical vertebral numbers in late Pleistocene mammoths recovered from the North Sea is high (33.3%)…. Abnormal numbers were due to the presence of large cervical ribs on the seventh vertebra”.

This text can give the impression that the complete cervical spine was analyzed in each specimen i.e that the abnormal number of non-transitional vertebrae also was analyzed but the only type of numerical variation found was cervical ribs. This sentence must be clarified e.g “Here we show that the incidence of partial homeotic transformation in late Pleistocene mammoths recovered from the North Sea is high (33.3%)” and then delete Abnormal numbers were.. or/and revise the text further.

Methods section
L49: insert “mammoth” between “All” and “specimens” to clarify.

Figures
Figure 1 is very good and relevant to the content of the article.

I do only have improvement suggestions to Figure 1: The inset with the articulation facet in 1D) is great and clarifying to the readers. The other figures (A, B, C) would improve with the same type of inset that shows the articulation facet.
Also, for the convenience of the readers it could be clarified what is left/right in the picture and what is left/right on the specimen.

Experimental design

No comments.

Validity of the findings

Because of the fact that the development of cervical ribs has been documented in the literature as a sign of disturbance in early embryogenesis, I think the findings in this study are very important. Even if the sample size is very small and the fact that complete vertebral columns were not available.

Additional comments

This is a short but very interesting study of skeletal pathology in late Pleistocene mammoths. I recommend this manuscript for publication in PeerJ.

Suggestion to the Discussion section:
L101-115: A review about the relationship between the magnitude of inbreeding depression and environmental stress is useful to cite somewhere in the text, for example: Armbruster P, Reed DH. 2005. Inbreeding depression in benign and stressful environments. Heredity 95:235–242.

·

Basic reporting

No Comments

Experimental design

No Comments

Validity of the findings

No Comments

Additional comments

A few formal proposals:

In the headline the provenance of the fossils (North Sea) and the species (Mammuthus primigenius) should be mentioned.

The Late Pleistocene is defined as a chronostratigraphic unit. Therefore a capital letter should be consequently used for “Late”.

Line 76: africana instead of “Africana”.

Line 82: The statement is generalized to all mammoths. It should be related to the number of the examined fossils and the area of their origins.

Figure: The figured vertebrae should be isolated from their inhomogeneous backgrounds. Probably it would be helpful to figure the bones in scale to each other and to add just one graphical scale. The arrows should be explained in the caption.

---

## Round 0.2 · accepted · Accept

I noted two minor typos that should be fixed during production:

line 28: remove extra semicolon in reference list
line 99: "tobe" needs space in it